# Review of Visual Simultaneous Localization and Mapping Based on Deep Learning

**Yao Zhang, Yiquan Wu \*, Kang Tong, Huixian Chen and Yubin Yuan**

College of Electronic and Information Engineering, Nanjing University of Aeronautics and Astronautics, Nanjing 211106, China; yaozhang_nuaa@nuaa.edu.cn (Y.Z.); tkangcv@nuaa.edu.cn (K.T.); chenhuixian2021@nuaa.edu.cn (H.C.); harley_yuan@nuaa.edu.cn (Y.Y.)

\* Correspondence: imagestrong@nuaa.edu.cn; Tel.: +86-137-7666-7415

**Abstract:** Due to the limitations of LiDAR, such as its high cost, short service life and massive volume, visual sensors with their lightweight and low cost are attracting more and more attention and becoming a research hotspot. As the hardware computation power and deep learning develop by leaps and bounds, new methods and ideas for dealing with visual simultaneous localization and mapping (VSLAM) problems have emerged. This paper systematically reviews the VSLAM methods based on deep learning. We briefly review the development process of VSLAM and introduce its fundamental principles and framework. Then, we focus on the integration of deep learning and VSLAM from three aspects: visual odometry (VO), loop closure detection, and mapping. We summarize and analyze the contribution and weakness of each algorithm in detail. In addition, we also provide a summary of widely used datasets and evaluation metrics. Finally, we discuss the open problems and future directions of combining VSLAM with deep learning.

**Keywords:** simultaneous localization and mapping; machine vision; deep learning; visual odometry; loop closure detection; mapping

## 1. Introduction

Simultaneous localization and mapping (SLAM) refers to a technology that enables robots to achieve self-localization and mapping while moving in an unexplored environment. In the initial stage, SLAM relied heavily on sonar and LiDAR sensors, which provide a high level of accuracy but come with considerable weight, high costs, and fragility. In contrast, visual sensors are lightweight, affordable, and easily deployable, and thus became an alternative. Visual simultaneous localization and mapping (VSLAM) can utilize visual sensors that work as human eyes to perceive the surrounding environment and gain rich environmental information, facilitating positioning and navigation in complicated real-world environments. VSLAM technology plays a significant role in several fields, including intelligent robots, autonomous vehicles, drones, unmanned vessels, military rovers, augmented reality (AR) and virtual reality (VR). In the intelligent factory, for example, the robots automatically sort and transport the goods, achieving a completely automated production line. In special scenarios where the global positioning system (GPS) fails (such as caves, tunnels, deep water, electromagnetic interference, etc.), self-driving vehicles, rescue robots, and unmanned underwater vehicles can rely on VSLAM to complete the rescue, detection, and long-distance cruising. In hazardous military areas, military detection vehicles can carry out high-risk tasks, including reconnaissance, demining, and hazardous waste disposal. Additionally, the emerging AR and VR technologies in recent years interweave virtual cyberspace with the physical world. The 3D map reconstructed by VSLAM can provide geometric information about the scene, and render the superimposed virtual objects to maintain their geometric consistency with the real world, making the virtual space more realistic. As the demand for VSLAM technology surges, various novel methods and technologies are emerging, making it a hot spot in research.

The concept of SLAM was initially presented at the IEEE Robotics and Automation Conference in 1986. Smith et al. [1] proposed the application of the estimation theory-based method for the localization and mapping of robots, marking the integration of robotics with artificial intelligence. SLAM has since developed over three decades. The first research on visual navigation was carried out by Ayache and Faugeras [2] in 1988, while Crowley [3] investigated Kalman filter-based visual navigation for mobile robots. Prior to 2000, filter-based methods were commonly used in SLAM. Although some excellent papers, such as PTAM [4], used bundle adjustment (BA) optimization for VSLAM, the error of nonlinear optimization and high computational load made it unpopular among researchers. In 2009, the sparsity of BA was discovered [5], which led to the rise of graph optimization-based methods such as DTAM [6], LSD-SLAM [7], SVO [8], ORB-SLAM [9] and other outstanding algorithms. With the rapid development of artificial intelligence technology and deep learning, researchers began to explore how to apply deep learning to traditional filtering and graph optimization-based methods to address the inevitable environmental adaptability problems of traditional filtering methods. Consequently, employing deep learning to resolve SLAM-related problems has emerged as a prevalent trend.

Currently, VSLAM methods can be broadly categorized into three classes: filter-based, optimization-based, and deep learning-based methods. The filter-based methods, including Kalman filter (KF), extended Kalman filter (EKF), unscented Kalman filter (UKF), and particle filter (PF), are not capable of handling large-scale outdoor scenes. As the requirements of people for VSLAM technology increase, optimization-based methods have emerged, mainly based on graph optimization. In recent years, the application of deep learning in machine vision has achieved greater improvements in operation efficiency, performance precision, and robustness. There is growing interest in using data-driven learning methods to resolve VSLAM problems.

The advantages of deep learning-based methods can be listed as follows:

1.  More powerful generalization ability. Compared with hand-crafted features, deep neural networks can automatically extract the features relevant to tasks, take full advantage of rich image information, and adapt better to complex scenes, such as motion blur, dynamic environment and large-scale scenes, that were previously difficult to handle with traditional algorithms.
2.  Advanced semantic information. Deep learning can extract more advanced semantic information, which can aid in the construction of semantic SLAM and the understanding of scene semantics. Furthermore, learning-based methods are more prone to establishing a connection between abstract elements (such as semantic labels) and understandable terms, which is difficult to achieve using mathematical theories.
3.  The data-driven form. Deep learning is a data-driven approach, which is more in line with the form of human–environment interaction and has greater development space and research prospects.
4.  The capability of fully exploiting vast amounts of sensor data and the hardware's computational power. As the VSLAM system operates, a significant amount of data generated by the sensors and optimization parameters of the neural network result in a substantial computational load. Fortunately, advances in hardware computing power have provided a reliable solution for maintaining real-time performance. Deep learning-based VSLAM methods can be optimized for parallel computing and large-scale deployment, which can lead to faster processing speed and lower power consumption.
5.  Learning from past experience. Deep learning-based methods can continuously enhance their models by drawing on past experience. By building generic network models, learning-based methods can automatically train and discover new solutions when faced with new scenarios, further improving their models.

As VSLAM has exploded in popularity, several reviews have been published in re-cent years. However, these reviews have not been systematic, comprehensive, and in-depth enough. Chen et al. [10] proposed a taxonomy of the vslam algorithms and classified these algorithms, but did not include the latest progress. Debeunne et al. [11] focused on

solutions for the fusion of vision SLAM and LiDAR SLAM from the perspective of the 6G wireless network. Huang et al. [12] summarized the technology of VSLAM, LiDAR SLAM, and their fusion, covering topics such as sensor products, open-source systems, multi-sensor calibration and the fusion of hardware, data, and task layer. Jia et al. [13] provided details on the classical structure of VSLAM as well as the latest algorithms, but did not focus on deep learning as the main part. Chen et al. [14] focused on semantic information and summarized the outstanding research on combining semantic information and traditional VSLAM for localization and mapping.

Our contributions can be outlined as follows:

1. Systematic review of deep learning-based methods for VSLAM. We make a comprehensive review of deep learning-based methods for VSLAM.
2. In-depth analysis and summary of each algorithm. We summarize and analyze the significant deep learning approaches applied in VSLAM, and deeply discuss the contribution and weakness of each approach.
3. A summary of widely used datasets and evaluation metrics. We list the widely used datasets and evaluation metrics for VSLAM. For the convenience of readers, we have also provided the link to each dataset.
4. Discussion of the open problems and future directions. We comprehensively discuss the existing obstacles and challenges to deep learning-based VSLAM and indicate the potential development direction for future research.

We provide an overview of the latest VSLAM methods based on deep learning in recent years, aiming to assist researchers in related fields to better understand the research status and development prospects of deep learning-based VSLAM. This paper is organized as follows. In Section 2, we briefly review the development process and basic framework of VSLAM. In Sections 3–5, we comprehensively summarize the deep learning-based VSLAM method from three aspects: visual odometry (VO), loop closure detection, and mapping. We analyze the strengths and limitations of each algorithm in detail. In Section 6, we provide a summary of widely used datasets and evaluation metrics. Based on existing work, we discuss the open problems and future directions in Section 7. Finally, we conclude our review in Section 8.

## 2. Key Structure of SLAM System

Similar to the classical SLAM system, VSLAM system consists of four primary components: the frontend, the backend, loop closure detection and mapping. The frontend is mainly concerned with the motion relations of the camera, utilizing adjacent frames to estimate the camera pose and reconstruct a local map. The role of the backend is to optimize the results obtained from the frontend, which obtain optimal positional estimates and more accurate maps via a filter or nonlinear optimization-based methods. Loop closure detection is used to determine whether or not the robot has returned to its previous position. According to the results of loop closure detection, the frontend can update the trajectory and map while eliminating the error accumulation to obtain a global consistent trajectory and map. The mapping aims to construct a map required for navigation, positioning, and other tasks.

### 2.1. Frontend

The frontend obtains image frames as the camera moves while calculating the changes of camera pose between adjacent frames. Since the inter-frame estimation only considers adjacent frames and the motion error between two images is inevitable, this error will accumulate after multiple transmissions, resulting in trajectory drift. Therefore, backend optimization and loop closure detection are necessary for eliminating error accumulation. The main methods of frontend can be classified into the feature-based method and direct method.

The feature-based method relies on feature points. The commonly used feature points include scale-invariant feature transform (SIFT) [15], speeded up robust features (SURF) [16] and oriented FAST and rotated BRIEF (ORB) [17]. SIFT is invariant to rotation, scaling and

luminance changes, which makes it highly significant and informative. It is suitable for fast and accurate matching in massive databases with minimal misidentification. However, the large number of floating-point operations required by SIFT makes it challenging to process in real-time. On the other hand, SURF is an optimized version of SIFT. SURF is based on the Hessian matrix for feature point detection and uses box filters to simplify the two-dimensional Gaussian filter when constructing the scale space. It greatly accelerates the detection speed and provides better stability. Meanwhile, ORB combines Features from accelerated segment test (FAST) [18] feature point and Binary robust independent elementary features (BRIEF) [19] feature descriptor. The FAST feature point offers fast operation speed but lacks both scale invariance and rotation invariance. ORB improves the original BRIEF to achieve the combination of the speed of FAST and both scale invariance and rotation invariance, making it able to be processed in real-time. Nonetheless, ORB is very sensitive to feature missing, and has weak robustness in environments with weak or no texture.

After feature points are detected, the next step is feature matching. When there is significant motion and appearance change between adjacent frames, calculating the distance between descriptors, such as Hamming distance and Euclidean distance, is necessary for feature point matching. The commonly used methods of matching include brute force, cross and K-nearest neighbor (KNN). Additionally, optical flow tracking is suitable for a situation with minor changes, whereby the motion relationship of camera is obtained by tracking key points to minimize the photometric error between two consecutive frames. The operation of optical flow tracking is fast because the descriptor is not necessary. However, it follows the gray-scale invariant assumption and is extremely sensitive to luminosity changes. In the case of significant luminosity changes, tracking loss is prone to occur.

The direct method does not rely on feature points or descriptors for motion estimation as it directly computes the motion relationship between two frames based on pixel-value information. The optimization of the camera pose is achieved by minimizing the photometric error function. So, it is particularly robust in environments with weak or no texture. However, like the feature-point method, the direct method is greatly affected by lighting variations and requires slow camera movements or a high sampling frequency. Additionally, the feature point method suffers from the loss of potentially valuable image information due to the involvement of only a few pixels. The non-convexity of the image also makes the optimization of the direct method prone to falling into a local optimum.

*2.2. Backend*

Backend optimization is essentially a state estimation problem, consisting of two methods: the filter method and the BA method.

The early backend optimization frameworks mainly utilized filters. In 1960, Kalman [20] proposed a recursive algorithm for discrete systems to achieve linear filtering, which was applicable to a linear system with Gaussian-distributed noises. However, in practice, a linear mathematical model is often insufficient for describing system characteristics, and many systems exhibit different degrees of nonlinearity. To account for this, the extended Kalman filter (EKF) [21] is proposed. The performance of EKF depends heavily on the degree of local nonlinearization and the original uncertainty. When a system is strongly nonlinear, EKF violates the assumption of local linearity and neglects the higher-order terms, which may result in significant error and divergence.

Furthermore, as the operation of the system continues, the data that need to be updated and maintained, as well as the size of the co-variance matrix, increase exponentially. This not only occupies a large amount of storage space but also renders the EKF unsuitable for large-scale environments. Ullah et al. [22] apply linear the Kalman filter (KF) and EKF SLAM in mobile robots. These algorithms are tested via simulations, which demonstrate their accuracy and viability for locating landmarks and mobile robots. Based on the idea of deterministic sampling, Julier et al. [23] proposed the unscented Kalman filter (UKF), which combines the unscented transform (UT) with the Kalman filter. Unlike the EKF, the UKF

avoids the complex Jacobian matrix and does not increase the difficulty of implementation because of a more complex system model, and the higher-order terms are not ignored.

Particle filter (PF) [24] is another commonly used filter method that employs the Monte Carlo simulation approach to achieve optimal estimation in the Bayesian filtering framework. PF obtains a set of continuously updated particles in the state space by importance sampling, where the particles with smaller weights are discarded, and those with larger weights are replicated to make the particle distribution closer to the practical state probability distribution. However, resampling leads to more offspring of particles with larger weights and fewer or even no offspring of particles with smaller weights, which is prone to a lack of sample validity and diversity, as well as the problem of sample impoverishment [25]. Thus, it is equally unsuitable for application in large-scale environments.

The filter methods are not suitable for large-scale SLAM because of their limitations. Additionally, the Markov property of filters makes subsequent loop closure detection difficult to deal with. Furthermore, the filter methods only use part of the historical data, while the nonlinear optimization method uses all data, theoretically leading to better performance. Through a series of Monte Carlo experiments, Strasdat et al. [26] demonstrate that the nonlinear method is better than the filter-based method in unit computing time.

The nonlinear method minimizes a cost function, which is constructed based on the error of the observed value and estimated value, to make the model function more closely to the actual model. Since the cost function is always nonlinear, it needs to be solved via least squares, which is what we call BA [27]. The general idea of BA is to solve the corresponding least squares problems, which are constructed according to pose estimation and spatial point position. Commonly used methods include the Gauss–Newton algorithm, gradient descent, and Levenberg–Marquart.

No matter which method is used, the final results of the incremental equation are in the form of $H\Delta x = g$ or $\left(H\lambda D^T D\right)\Delta x = g$. For complicated models, calculating the $H$ matrix is challenging. The consensus amongst scholars is that the heavy computational burden associated with BA optimizing renders it impractical for real-time applications. With the increased recognition of the sparse structure of the $H$ matrix [5], this issue is resolved. In contrast to the Markov assumption employed in filter-based algorithms, which only consider two adjacent states, nonlinear optimization utilizes all previous states and benefits from all historical information. Moreover, the sparsity of the state matrix also makes the computation faster and loop closure detection easier.

## *2.3. Loop Closure Detection*

Backend optimization can correct the maximum error of frontend results, but the cumulative error always exists. Loop closure detection can eliminate the influence of the cumulative error, which plays a significant role in system precision and the global consistency map.

Bag-of-words (BoW) [28] is a widely used method for loop closure detection. It generates dictionaries by clustering visual features (such as SIFT and SURF) in images and determines loop closure by calculating the similarity of dictionaries across keyframes. Furthermore, the bag-of-words model is essentially an unsupervised machine learning process, and the computation of similarity between two frames can essentially be seen as a classification problem; researchers have begun to explore how to integrate deep learning with loop closure detection. In recent studies, Li et al. [29] replaced the relevant structures in ORB-SLAM with various feature point extraction and matching networks of deep learning (Superpoint, D2NET, HF-NET, etc.) for relocation and loop closure detection. Gao et al. [30] proposed a multilayer neural network based on stacked denoising autoencoders (SDAs) to learn image features from original image data in an unsupervised manner. These studies demonstrate the trend toward integrating deep learning into loop closure detection as part of the continued development of SLAM technology.

## 2.4. Map Types

The purpose of mapping is to simultaneously achieve self-localization and construct a map of the unknown environment. This map should accurately describe the environment's characteristics while minimizing the complexity of the map to ensure real-time performance. Therefore, the representation of the map should be selected according to the specific environmental requirements and actual tasks. In terms of dimensions, the map's representation can be divided into two-dimensional and three-dimensional. Two-dimensional maps include the geometric map, grid map and topological map [31].

The geometric map consists of a series of geometric features (such as points, line segments and curves), which offers high precision and requires little computation. However, a large amount of perceptual data is needed for large-scale environments [32]. A grid map divides the map into equal-sized grids with three states, occupied, free, and unknown, to distinguish passable areas from obstacle areas. It is widely used for path planning and navigation but may suffer from inadequate real-time performance in either large-scale environments or excessively detailed grid subdivisions. Additionally, the topological map models the environment as a graph, which is composed of the nodes and the edges between nodes, with the nodes corresponding to the places in the real environment and the edges corresponding to the connection between different places. This map offers a more abstract representation of the environment and has advantages in path planning and navigation, particularly for larger-scale environments.

Among the 3D maps, the point cloud map is the most widely used. Although it retains detailed information about the original environment, it is generally large-scale and takes up a lot of memory for information which is unnecessary for tasks. Based on the octree structure, octomap [33] is a compressed 3D grid map, with less ambiguity compared to the 2D grid map and less memory compared to point cloud map. However, it is hard for real-time path planning and searching because of its complexity. Other types of maps, such as feature maps, the ESDF map [34], TSDF map [35], semantic map [36], offer further options for mapping and analysis.

Different map types vary in memory usage, complexity and properties, thereby rendering them suitable for specific tasks. We present a summary of their properties and frequently employed tasks in Table 1. To enhance the readers' perceptual understanding of diverse maps, we provide a visualization of different maps in Figure 1.

**Table 1.** Comparison of different map types.

| Map Type | Memory | Complexity | Localization | Navigation |
|:---:|:---:|:---:|:---:|:---:|
| Grid map | High | Medium | √ | √ |
| Topological map | Low | Low | √ | |
| Feature map | Low | Low | | √ |
| Semantic map | Medium | Medium | √ | √ |
| Point cloud map | High | High | √ | |
| Octomap | Medium | High | | √ |
| ESDF map | Medium | High | √ | √ |
| TSDF map | Medium | Medium | √ | √ |

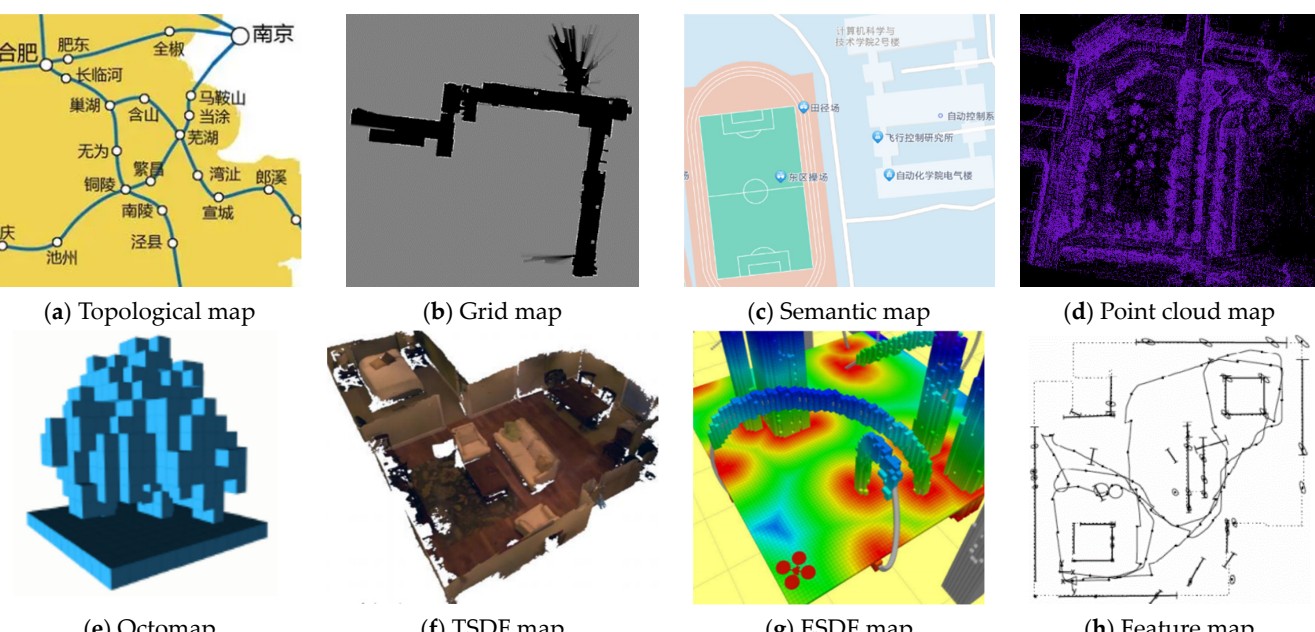

**Figure 1.** The visualization of different maps.

## 3. Visual Odometry (VO) with Deep Learning

Combining deep learning with VSLAM has been a general trend; the common practice is to replace one or more modules and steps of VSLAM with those of deep learning methods. In this paper, we provide a comprehensive overview of the combination of deep learning and different components of VSLAM, including visual odometry (VO), loop closure detection and mapping.

Visual odometry tracks and calculates two adjacent frames to obtain the camera pose, depth map and local consistent trajectory. The feature extraction step can be re-placed with a deep neural network. Conventionally, feature detection and extraction are performed using handcrafted features, which is time-consuming and extremely dependent on prior domain knowledge. Conversely, deep learning methods automatically extract the features required for tasks through adopting the highly expressive deep neural net-work as a universal approximator. That is why deep learning models are more adaptable to various environments, especially because the situation is hard to model manually, such as in instances of a dynamic environment, complex scene and motion blur.

### 3.1. Feature Extraction

Employing deep neural networks to extract features is a common operation in the VSLAM system. DeTone et al. [37] proposed a self-supervised framework that is suitable for multi-view geometry problems. SuperPoint consists of a shared encoder and two decoders. The VGG-style shared encoder is used to reduce the dimension of input images. The output of encoder is shared by the detector decoder and descriptor decoder. The trained interest point detector decoder, called MagicPoint [38], generates the pseudo-ground truth interest points. However, when the viewpoint changes, many potential interest points are missed. To overcome this limitation, homographic adaptation is introduced, which warps the input image multiple times to help MagicPoint obtain the potential interest points from different viewpoints and scales. The descriptor decoder is similar to the UCN and outputs a semi-dense descriptor, then performs bicubic interpolation and L2 normalization to generate a unit-length descriptor eventually. The computation and representation are shared between interest point detection and descriptor generation, and both are optimized simultaneously. Even though SuperPoint performs exceptionally well, the complex operations involved still cause time-consuming issues. To resolve this, MobileSP [39] proposes several algorithm and

hardware level design techniques through algorithm-hardware codesign, greatly improving processing speed while maintaining high precision.

For 3D projective geometry, Tang et al. [40] proposed a feature extraction algorithm GCNv2 and designed a binary descriptor that can easily replace ORB. The GCN [41] consists of an FCN with a resnet-50 backbone for feature extraction and a bidirectional re-current network for keypoint location. The large network architecture limits the application of GCNv2 to hardware platforms with limited resources, and the bidirectional recurrent network requires two or more frames to be input at the same time, increasing the computational complexity. Inspired by the idea of SuperPoint, GCNv2 performs prediction independently on a single image at a low resolution and greatly simplifies the network architecture. Furthermore, both the binarization of the feature vector and binary descriptor greatly speed up the matching. GCNv2 maintains high precision while improving computation efficiency. LIFT-SLAM [42] is a deep learning feature-based monocular system, which combines deep learning-based feature descriptors with the traditional geometry of VSLAM. The learned invariant feature transform (LIFT) network consists of a detector, orientation estimator and descriptor which function independently of one another to extract local features, estimate orientations, and calculate descriptors, respectively. Original LIFT was trained on the phototourism image sets that differed from the VO dataset. LIFT-SLAM uses transfer learning to fine-tune the network with the VO dataset and improves the performance on cross-datasets. Additionally, an adaptive method is created to adjust the matching threshold according to the characteristics of different datasets.

Unlike the above methods that directly extract varying types of features, Xue et al. [43] proposed an end-to-end visual odometry framework to guide feature selection based on a deep convolutional dual-branch recurrent network. A context-aware feature selection mechanism was introduced in both spatial and temporary domains to enhance the ability of feature selection and distill motion-sensitive information for each branch. The model achieves great performance on KITTI and ICL_NUIM datasets and gains great generalization ability in various scenarios.

Most deep learning-based methods often rely on training datasets and struggle to adapt to unknown environments, sometimes even at the cost of efficiency for increased precision. DF-SLAM [44] designs local feature descriptors to enhance the precision of data association between frames, effectively replacing handcrafted descriptors. Although deep neural networks can improve performance with more layers, they can also be time-consuming. DF-SLAM adopts TFeat with a triplet network. Borrowing from the hard negative mining strategy of HardNet, it combines the hard negative mining strategy with TFeat to improve the model performance. The result shows that it not only improves efficiency and stability but also strong robustness in challenging scenes, such as intense illumination changes. Additionally, the shallow network architecture of TFeat ensures real-time performance.

Furthermore, the static feature information in the image, dynamic features also merit our consideration. You only look once (YOLO) is the most classical target detection network, which can detect video or images in real-time with high accuracy. Soares et al. [45] employed YOLOv1 and Mask-RCNN for object detection and instance segmentation separately. Once a person appears in the image, the object detection switches to instance segmentation and then removes the key points inside the people. Otherwise, only object detection is active. However, this work only focuses on the people and neglects other dynamic objects. Based on the LeGO-LOAM, Kim et al. [46] adopted YOLOv3 to multi-target recognition and described the position of targets on the reconstructed indoor 3D map. Wu et al. [47] proposed a lightweight network version of YOLOv3 by replacing the original darknet-53 backbone network with darknet-19, achieving high-speed detection. Compared with YOLOv3, YOLOv4 replaces the backbone network from Darknet53 to CSP Darknet53 and introduces spatial pyramidal pooling to better extract features at various scales. Bala et al. [48] used YOLOv4 to detect objects, and the input image features were updated with the ORB features extracted from the objects. Object detection inevitably suffers from the

problems of missed and false detections. To mitigate these issues, Wang et al. [49] incorporated YOLOv5n object detection with a leak detection judgment and repair algorithm, which effectively decreased the influence of dynamic features.

In the VO problems, the difference in geometric feature information across frames is of more concern. The introduction of an efficient attention mechanism in the network enhances the representational capacity of model and raises the weight of the feature region, learning to feature information relevant to tasks more easily. Attention-SLAM [50] simulated the navigation behavior pattern of humans and proposed a saliency model, SalNavNet, to predict the salient areas with different degrees of importance and adopt them as weights. Differently from the traditional BA methods, a weighted BA method is proposed to pay more attention to feature points of salient areas and reduce the trajectory error effectively. Additionally, an adaptive exponential moving average (EMA) module is introduced to mitigate the center bias problem.

Most deep leaning-based methods directly extract features from original images. SuperPoint [37] introduces homographic adaptation to help the network learn from the images that captured from different viewpoints, which make it more suitable for multi-view geometry problems. The binary features designed by GCNv2 [40] can easily replace the traditional ORB features, which lead to an acceleration of subsequent feature processing. Due to using SIFT as the supervise signal, LIFT [42] inherits the rotation and scale invariance of SIFT, and deep learning makes it exhibit powerful robustness across datasets. The feature selection mechanism [43] can select the most suitable feature type based on the current scene, and applying it to different types of features may be a good idea, such as the three features mentioned above. Additionally, the attention mechanism [50] makes the feature regions more significant and gains more precise features. However, it is not common to add attention modules to a feature extraction network. That is because weighting the entire image requires a large amount of computation, which is time-consuming. To deal with dynamic features, many efforts [45–49] utilize YOLO to detect objects, some of which accelerate inference speed by replacing lightweight backbone networks [47], obtain more accurate dynamic information combined with instance segmentation [45] and repair leak detection [49].

Employing deep neural networks to extract image features has become a well-established method, as these deep features exhibit superior generalization and robustness compared to traditional manual features. The network architecture design becomes more lightweight to ensure real-time performance. To accelerate subsequent feature processing, the feature designed in binary form and hardware algorithm optimization are both available solutions. A summary of feature extraction algorithms in Table 2 is presented.

It can be seen from Table 2 that most feature extraction methods are supervised and focus on indoor scenarios. However, outdoor scenarios present greater challenges due to various weather conditions, environmental settings, and dynamic objects, which can significantly affect the quality of extracted features, including motion blur, occlusion, and changes in lighting. In light of this, several methods employ different networks, such as GAN, recurrent network, YOLO, and Mask-RCNN, to design features that are specific to particular scenarios and datasets. The huge architecture results in the time-consuming issue, and some studies aim to accelerate the inference speed through various solutions, including replacing lightweight backbone networks, compressing network parameters and only considering a single category of features. To some extent, although some methods achieve real-time performance, sometimes precision is sacrificed for a faster inference speed. In addition, most methods have not yet been tested on real-world robots, suggesting a limitation in the practical application of these approaches.

**Table 2.** A summary of the feature extraction.

| Method | Supervision | Scenario | Contributions | Weakness |
|---|---|---|---|---|
| [37] | Self-supervised | Indoor, Outdoor | homographic adaptation pseudo-ground truth | the high repetition rate of feature points it does not learn from actual data |
| [42] | Supervised | Urban, Indoor | transfer learning Learned invariant feature transform (LIFT) | the performance will become poor when loop closure cannot be detected |
| [43] | | Urban, Indoor, Highways | dual-branch recurrent network context-aware guidance mechanism used for feature selection | could not run in real-time |
| [45] | | human populated | YOLOv1 for dynamic object detection Mask-RCNN instance segmentation | only the category of person was considered |
| [48] | | Road | YOLOv4 for object detection | it has not been tested in real world |
| [40] | | Indoor | GCNv2 simplified network binary descriptor vector as the ORB feature eigenvector binarization accelerates training | it predicts projective geometry and not generic feature matching only considers indoor scenarios |
| [44] | | | very lightweight network framework robustness of intense illumination changes can run in real time | fewer comparison algorithms |
| [46] | | | YOLOv3 for object detection lightweight network Darknet19-YOLOv3 | only single object detection |
| [47] | | | a depth-RANSAC screening method | performance degradation because of the camera rotation |
| [49] | | | YOLOv5n for object detection a leak detection judgment and repair algorithm | it sacrifices some of the accuracy for the advantage of speed |
| [50] | | | visual saliency model (SalNavNet) exponential moving average module | poor performance under motion blur and rapid rotation |

### 3.2. Motion Estimation

The depth information of objects in an image can be used to optimize the pose. However, a monocular VSLAM system cannot recover the depth information for a single image. Generally, the traditional methods gain the depth information of map points via triangulation, inverse depth, time of flight (TOF) and structured light. Despite the advancements made in depth estimation, there are still limitations associated with current methods. Fortunately, deep learning techniques have seen immense popularity and show great potential in promoting further development in this field.

Zhou et al. [51] proposed a completely unsupervised end-to-end framework, namely SFMLearner, which includes depth, pose and explainability modules. The single-view depth prediction adopts the dispnet [52] architecture and outputs the depth maps of input images. The pose module takes both the target view and the nearby view as input and predicts the 6-DoF relative pose. The supervision signal for depth and pose modules is generated by synthesizing source images from different camera poses. The explainability module predicts multi-scale explainability masks for the corresponding source–target pair. On the other hand, for dynamic scenes, occlusion, and non-Lambertian surfaces, gradient corruption easily occurs, leading to training failure.

Differently from SFMLeaner directly generated a depth map, Godard [53] regarded depth estimation as an image reconstruction problem and proposed an unsupervised depth estimation network architecture to generate disparity images which are used to rebuild pixel depth. A depth estimation training loss with an inbuilt left–right consistency check is introduced to enforce consistency between the predicted depth maps during training, improving the robustness and performance of the system. GeoNet [54] is a jointly unsupervised learning framework for depth, optical flow and pose, adopting a divide-and-conquer strategy to solve the scene's rigid flow and object motion adaptively. The network architecture is composed of DepthNet, PoseNet and ResFlowNet. The rigid structure reconstructor consists of DepthNet and PoseNet, aiming to reconstruct the rigid scene's structure. ResFlowNet is also called a non-rigid motion localizer which is used to learn the residual non-rigid flow and acts as compensation for dynamic objects. For occlusions and non-Lambertian surfaces, an adaptive geometric consistency loss is introduced which consists of a forward–backward consistency check and left–right consistency check. The

experiment result shows that GeoNet performs well in texture-ambiguous regions and shaded dim areas.

Similarly to SFMLearner and GeoNet, D3VO [55] proposed a self-supervised depth estimation network that tightly incorporates depth, pose and uncertainty. This self-supervised network consists of depthnet and posenet and they are bridged by minimizing the photometric error originating from both static stereo warping and temporal warping. It also predicts the brightness transformation parameters to solve the problem of inconsistent illumination between training image pairs. However, only modeling brightness change is not sufficient to solve all violations of the photometric constancy assumption. Therefore, it also predicts the photometric uncertainties based on the distribution over the possible brightness values of each pixel to improve the accuracy of depth estimation and provide a learned weighting function for the photometric residuals.

The three methods mentioned above leverage different networks and strategies to consider depth, pose and uncertainty. Unlike utilizing CNN to generate depth information directly, GANVO [56] firstly combined the generative adversarial network (GAN) with the recurrent unsupervised learning approach to jointly estimate pose and depth. The depth network was designed based on GAN and consists of a generator network, a discriminator network and an encoder network. The input image is mapped to the feature vector by the encoder network. The generator network maps the feature vectors to the depth images. The output depth image is mapped to the color space and the discriminator distinguishes this mapping between the original image and the output of the generator. The pose network outputs the 6-DoF relative pose after a CNN and two LSTM modules. SGANVO [57] firstly applied stacked GAN for self-motion and depth estimation. The network structure of each layer is similar to that of GANVO, including generator and discriminator, and the convLSTM is added to connect each layer.

Differently from the methods that require a priori information, the a priori object model is unnecessary for CubeSLAM [58], which extends the 3D object detection of a single image to multi-view object SLAM to jointly optimize the object and camera pose. This method adopts YOLOv2 for indoor scenarios and MS-CNN for outdoor scenarios to perform 2D object detection and generate 2D bounding boxes. According to the 2D bounding box based on the vanishing point (VP), the intersection of parallel lines after projection onto perspective images, high-quality cuboid proposals are generated. CubeSLAM proposes different cost functions to score them. Additionally, an object association method is proposed for the occlusion and dynamic. To jointly optimize the pose of cameras, objects and points, the multi-view BA with new object measurements is proposed. MonoRec [59] is a semi-supervised reconstruction architecture that consists of a MaskModule and a DepthModule. Based on the structural similarity index measure (SSIM) instead of the sum of absolute differences (SAD), the information from multiple consecutive images is used to construct a cost volume. The MaskModule predicts moving object masks according to the photometric inconsistencies encoded in the cost volumes to indicate the possibility of pixels belonging to moving objects. By incorporating moving object masks and cost volumes, DepthModule outputs the accurate depth map of static and dynamic objects. Shamwell et al. [60] proposed an online error correction (OEC) module and an unsupervised deep neural network, namely visual–inertial odometry learner (VIOLearner). The OEC module is used to correct intermediate output errors of a network. VIOLearner performs convolution on Jacobian functions at multiple spatial scales for multi-step trajectory estimation and generates a trajectory estimator to correct the trajectory online.

The images captured by RGB-D cameras come with depth information and make pose estimation easier. The dynamic objects in the scenes will impact the performance of the system. DDL-SLAM [61] is an RGB-D SLAM framework used to decrease the influence of moving objects on the pose estimate. DUNet is adopted for semantic segmentation and a multi-view geometry algorithm is introduced to filter out the dynamic content of a scenario. The obscured background of the current frame will be reconstructed, leveraging the static information of prior frames. These synthesized RGB frames without dynamic

objects and their corresponding depth maps are used to generate the local point cloud. Although DDL-SLAM performs more accurately and robustly in highly dynamic scenarios, the problem of real-time performance requires improvement.

Similarly to DynaSLAM [62], DynaSLAM II [63] adopts Mask-RCNN to perform semantic segmentation as a priori semantic information. Differently from DynaSLAM, which ignores dynamic object information, DynaSLAM II leverages different dynamic objects in the scene to perform multi-objective tracking. The final experiments show that multi-objective tracking benefits motion estimation and not only provides rich information for scenario understanding but also improves localization and trajectory precision. WF-SLAM [64] adopts Mask-RCNN [65] on original RGB-D images to implement semantic segmentation and generate semantic masks. However, it is hard to detect objects which are semantically static but actually dynamic, such as a book shaking in hands. Therefore, it combines epipolar geometry constraints and semantic segmentation to accurately detect dynamic objects and remove dynamic feature points. At the same time, a weighted optimization algorithm is proposed to assign different weights to static and dynamic feature points and jointly optimize the pose and weights. Although WF-SLAM obtains great localization precision and robustness in dynamic scenarios, it cannot run in real time.

Some frameworks jointly consider depth, pose and uncertainty. SFMLearner [51] directly generates a depth and pose through neural networks, but it fails for occlusion and non-Lambertian surfaces. Godard et al. [53] introduce a left–right consistency check to improve the reliability of the predicted depth. Inspired by the above two methods, GeoNet [54] can deal with occlusion and non-Lambertian surfaces through combining the a scene's rigid flow and object motion and introducing a forward–backward consistency check and left–right consistency check. D3VO [55] takes photometric uncertainty into account, improving the predicted depth of images with inhomogeneous brightness. Additionally, there are several novel methods for improving motion estimation. CubeSLAM [58] and MonoRec [59] build a universal representation and evaluate it to remove incorrect movement relationships. The OEC module integrates the IMU data and corrects the absolute trajectory estimation based on hypothesis trajectories. It has good generalizability under motion, image perturbations and domain shifts. GANVO [56] employs GAN to produce accurate depth and camera motion estimates which demonstrates improved performance in terms of accuracy and robustness in challenging environments such as low-texture, fast-motion environments, and occlusions. The stacked GAN has demonstrated that within a certain range, deeper networks have stronger performance [57]. For dynamic scenes, many efforts utilize various segmentation networks to segment dynamic objects. DDL-SLAM [61] employs a lightweight segmentation network but it cannot run in real time because of the complex subsequent multi-view geometric algorithm. WF-SLAM [64] integrates epipolar geometry constraints and semantic segmentation, allowing it to detect potential dynamic objects. In contrast to the two methods which simply delete dynamic elements, DynaSLAM II [63] tracks multiple dynamic objects to improve the trajectory.

Depth information is always crucial for motion estimation. Although depth information provided by RGB-D cameras simplifies the subsequent process, it is not always accurate and reliable owing to noise, occlusion, and non-Lambertian surfaces. It is necessary to utilize deep learning to correct or generate the depth information and achieve pose estimation and motion estimation. Table 3 summarizes and analyzes the above deep learning methods for motion estimation.

In Table 3, motion estimation mainly consists of depth, pose and uncertainty estimation. Some methods integrate various constraint to deal with challenging scenarios, such as left-right consistency, forward–backward consistency and photometric consistency. The semantic information reduces the impact of dynamic objects and also brings huge computation. It should be noted that most methods of motion estimation are unable to generalize to more datasets. There are many restrictions and prerequisites to employing these methods.

**Table 3.** A summary of motion estimation.

| Method | Supervision | Scenario | Contributions | Weakness |
|---|---|---|---|---|
| [51] | Unsupervised | Indoor, Urban | image reconstruction for supervising | easy to fail for dynamic scenes; occluded; non-Lambert surface artifacts on the boundary of the occluded area |
| [53] | | | single image depth estimation with left–right consistency | single-view datasets are not supported |
| [54] | | | forward–backward, left–right consistency; learns a scene's rigid flow and object motion | poor performance across datasets |
| [56] | | | GAN for pose and depth estimation without strict parameter adjustment | the performance on multiple datasets is not as good as that of other supervised methods |
| [57] | | | stacked GANs a recurrent representation which can capture the temporal dynamic features. | is without loop closure detection with a certain drift |
| [60] | | Indoor, Room | online error correction (OEC) modules unsupervised learning | only a single source and target image are accepted cannot perform any type of BA adjustment |
| [58] | Supervised | Urban, Indoor | single-image 3D cuboid detection multi-view bundle adjustment | poor performance in some specific scenarios |
| [63] | | Outdoor, Indoor | cost-efficient bundle adjustment motion estimation and multi-object tracking are mutually beneficial | fewer comparison algorithms |
| [61] | | Indoor | DUNet semantic segmentation multi-view geometry weighted dynamic and static feature points | cannot run in real time |
| [64] | | Indoor | Mask-RCNN for a semantic mask epipolar geometry constraint | it is too time-consuming |
| [55] | Self-supervised | Indoor | integrates depth, pose and uncertainty temporal information is integrated into training | the generalization ability is not good, and there is a significant difference in performance across different datasets |
| [59] | Semi-supervised | Indoor, Road | structural similarity index measure (SSIM) maskmodule and a depthmodule | may fail when there are too many dynamic objects |

### 3.3. Selection of Keyframes

A video is composed of a series of image frames that appear at fixed intervals, and the adjacent frames are usually similar. When the camera moves slowly, there is too much unnecessary redundant information for the task in the image frames in a sequence, increasing a lot of unnecessary computation for the VSLAM task. Therefore, it is necessary to select one frame from a series of original image frames to represent a local frame, reducing the number of frames to be optimized. The common practice is to select keyframes based on the relative displacement and the number of tracked feature points, leading to the problem of a large amount of computation, poor real-time performance, and proneness to error selection. The key is how to efficiently select the valuable key frames while maintaining low computation and ensuring real-time performance.

Sheng et al. [66] proposed a deep neural network exclusively designed for keyframe selection and further proposed an end-to-end unsupervised framework to achieve keyframe selection and VO tasks at the same time, which is the first time two complementary tasks in a single framework were optimized. The keyframe selector has a two-stream structure which consists of a visual stream and a geometric stream. Both two streams share the same network architecture which originates from that of ResNet-18, but not network parameters. The visual feature and geometric feature which are extracted from two streams are fused through cross-modal attention, and the final similarity score is generated by combining the visual and geometric similarities. If the similarity score between the nearest keyframes and current frame is above a threshold, it will be inserted into the keyframe pool.

Inspired by the success of applying long short term memory (LSTM) to a structured prediction, Zhang et al. [67] proposed a LSTM-based supervised method, namely vsLSTM, which transformed keyframe selection into a structured prediction problem. LSTM is a recurrent neural network which does well in modeling long-range dependencies. vsLSTM

adopts bidirectional LSTM to model long-range dependencies in the forward and backward directions. The determinantal point processes (DPP) are used to enhance vsLSTM by modeling pairwise repulsiveness which aims to eliminate redundant frames in the selected frames. To sum up, vsLSTM predicts the important score of frames to select keyframes and DPP outputs the probability that different subsets of candidate keyframes are the keyframe sets. This work obtains great performance with sufficient annotated samples and also shows how to confront the challenge of a lack of annotated samples.

With regard to image quality and semantic information, Alonso et al. [68] proposed a keyframe selection algorithm that computes the image quality criterion and semantic score to evaluate original frames. The image quality criterion combines the blurriness score which is based on the Laplacian energy [69] and the brightness score which is based on the luminance of pixels. Additionally, semantic score is obtained based on semantic segmentation and achieved by MiniNet. Inspired by low-computation CNN frameworks, such as ERFNet [70], ENet [71], Deeplab-v3 [72], MiniNet is designed to run on CPU and consists of a down-sampling module, an up-sampling module and two convolutional branches which are concatenated between the down-sampling block and the up-sampling block, with a skip connection which is similar to that of Unet [73]. The ratio of image pixels that belong to the target class is taken as the semantic score, which is computed on the image of the MiniNet output with the same resolution as original image. The image quality criterion and semantic scores are combined to determine the best image as a keyframe.

For some time-consuming semantic segmentation networks, if each keyframe is segmented sequentially, new semantic information may not be available for the current frame during the tracking process, resulting in tracking failure. Considering this problem, Liu et al. [74] proposed a real-time visual dynamic SLAM algorithm based on ORB-SLAM3, namely RDS-SLAM. Based on semantic segmentation, it adopts a keyframe selection strategy with a bidirectional model instead of a sequential model, reducing the semantic delay to obtain the latest semantic information as much as possible and adapting to segmentation methods with different processing speeds.

The need for keyframes to be completely achieved by the deep neural network is unreasonable, and the integration of traditional geometric knowledge is necessary. Sheng et al. [66] combine visual and geometric similarities to select keyframes, which makes full use of the image information. In cases where geometric methods are insufficient, neural networks can be utilized to assess images and vice versa. Rather than increasing the number of types of image information, Zhang et al. [67] focus on the time of different frames. Bidirectional LSTM can model long-range dependencies in the forward and backward directions, leading to better keyframe candidates. The introduction of high-level semantic information contributes to the evaluation of the images. Alonso et al. [68] employ the lightweight MiniNet to calculate the semantic scores, so it can run in real-time. The architecture, complexity, and inference time of different segmentation networks vary. To adapt to semantic segmentation networks with different processing speeds, the bidirectional model greatly shortens semantic delay [74].

Selection of keyframes and loop closure detection share some similarities, as both require the extraction of image features and measurement of differences between images to identify images. In the feature extraction in Section 3.1, various types of algorithms are proposed to extract various features such as semantic, static and dynamic features. For the selection of keyframes, the crucial aspect is how to measure the degree of difference between image pairs, and loop closure detection puts more emphasis on the similarity. Table 4 summarizes and analyzes the above deep learning methods in terms of the selection of keyframes.

It can be seen at Table 4 that combining different types of features contribute to the selection of keyframes, such as semantic information, geometric features and deep features. Inspired by Section 3.1, employing YOLO to deal with the dynamic objects is a feasible solution. Taking into account the dynamic information may facilitate the similarity computation of keyframes.

**Table 4.** A summary of the selection of keyframes.

| Method | Supervision | Scenario | Contributions | Weakness |
|--------|-------------|----------|---------------|----------|
| [66] | Unsupervised | Urban, Indoor, | a keyframe selection network framework keyframe selection and VO are jointly optimized | the prediction of dynamic objects is challenging |
| [67] | Supervised | Urban, Outdoor | combines LSTM with the determinantal point process | poor performance in rapid-change scenarios |
| [68] | | Indoor | MiniNet for semantic segmentation image quality and semantic information | the segmented objects are individuals and it does not establish a connection between each individual |
| [74] | | | can process semantic segmentation methods with different speeds | it has not been deployed in a real robot system |

## 4. Loop Closure Detection with Deep Learning

Loop closure detection is vital for the relocation and elimination of accumulated errors. The general idea is to extract the feature information of the image frame and calculate the similarity between the current frame and the previous keyframes. The loop closure is considered detected when the similarity score reaches a certain threshold.

Gao et al. [30,75] adopted the stacked denoise autoencoder (SDA) to solve loop closure detection problem. SDA is an unsupervised multi-layer neural network, which consists of multiple denoise autoencoders (DA). Each DA layer is composed of an input layer, hidden layer and recovery layer, and the output of the current layer is the input of next layer. The final output feature response of SDA is used to calculate the similarity score. The similarity score of each pair of keyframes forms a similarity matrix to check the possible loops. Based on the denoise autoencoder, Merril et al. [76] rebuild HoG [77] descriptors to maintain illumination invariance. However, this work can only detect if the current position has been visited and cannot display keyframes matching the current frame.

There are always objects with different sizes in scenes, and features with multiple scales can contribute to detecting loop closure. Chen et al. [78] proposed a loop closure detection algorithm based on multi-scale deep feature fusion, which consists of a feature extraction layer, feature fusion layer and decision layer. The feature extraction layer adopts the pre-trained AlexNet [79] to extract deep features. However, if the size of an input image is unmatched by AlexNet, the input image needs to be cropped or compressed, resulting in information loss. Therefore, the feature fusion layer adopts spatial pyramid pooling (SPP) [80] to divide the feature map into blocks of different sizes at different scales. The features extracted from these blocks are combined to generate a fixed-size output and achieve multi-scale deep feature fusion. The decision layer performs a similarity calculation based on the output feature vectors of the images collected at the current time and the earlier time to detect loop closure. In addition, Chen adds weights to each feature node based on the distinguishability of the scene's feature nodes, improving the robustness under lighting changes and the accuracy of loop closure detection.

Memon et al. [81] combined deep learning with the super dictionary and proposed a novel loop closure detection method, which consists of deep CNN classifier, autoencoder, super dictionary and similarity detector. Based on the VGG16 [82] framework, the deep CNN classifier classifies objects as static or dynamic objects and extracts deep features from dynamic objects. The super dictionary and BoW [28] dictionary work together to ensure the speed of feature matching and reduce the risk of missing actual loop closures. The output feature vectors of CNN input the autoencoder and calculate the reconstruction error based on the mean squared error to detect if the current scene has already been visited or not. However, the current scene being visited does not mean that a loop closure is detected because many scenes are similar. The similarity detector will further check the visited scenes using the super dictionary and BoW dictionary to determine the final results.

An et al. [83] leveraged ResNet50 as the backbone of FCN and proposed an appearance-based loop closure detection method, which was named fast and incremental loop closure detection (FILD++). The global and local features are extracted via two passes of the

proposed FCN for filtering and reranking, respectively. An attention module is adopted to generate corresponding scores. Owing to the simplified network and sufficient strategies, FILD++ demonstrated good performance across different datasets and exhibited a speed advantage when applied on large datasets. To learn feature correlations effectively, Xu et al. [84] proposed ESA-VLAD, a novel network that employed EfficientNetB0 as its backbone and integrated a second-order attention module. The global features extracted from the network are used to retrieve the candidate images via hierarchical navigable small world (HNSW). Furthermore, an efficient geometrical consistency check based on local difference binary (LDB) descriptors is designed to verify matches. Zhang et al. [85] proposed a loop closure detection framework named AttentionNetVLAD which employed the R50-DELG network to simultaneously extract global and local features. The global features are used to select candidate frames via HNSW, while the local features are exploited for geometric verification between candidate pairs.

Differently from most works that directly use features extracted from CNN, Zhang et al. [86] preprocessed the CNN features which are extracted from the overfeat [87] network, including L2 normalization, PCA dimensionality reduction and whitening, to improve the representational capacity before carrying out the subsequent operations. Based on the idea of post-processing the features, after whitening, Wang et al. [88] compress feature information with selectable compression ratios to eliminate the redundant information of the dynamic environment. The temporal similarity constraint is added to reduce the similarity between adjacent images which are not loop closures.

Similarly to the selection of keyframes, Chen et al. [78] fuse the multi-scale deep features to calculate the similarity, which increases the number of feature types. Weighting each feature node when calculating similarity scores based on the illumination changes in the images also enhances robustness. In contrast to computing similarity scores based on image features, Gao et al. [30] model the problem as a stacked auto-encoder formulation. The scores are calculated by the output similarity matrix of the auto-encoder, which exhibits a stronger global perception ability for the entire images. However, as the image size increases, large computation is inevitable. FILD++ [83], ESA-VLAD [84] and AttentionNetVLAD [85] integrate the global and local features extracted by various networks to further filter image frames. Compared to methods that only consider global features, these methods have powerful generalization ability. Additionally, Memon et al. [81] integrate the autoencoder and super dictionary and detect loop closure based on dynamic objects features. Other efforts concentrate on the post-processing of CNN features [86] and information compression [88], which is dor reducing the processing time of each frame.

The integration of deep learning and loop closure detection primarily concentrates on feature extraction. Based on the similarity function and varying-scale feature information obtained via deep learning, these works calculate the similarity score between two frames and set a threshold to determine whether or not a loop closure is detected. Additionally, the illumination sensitivity and dynamic variation both affect the robustness of the VSLAM system. Thus, it is crucial to consider these factors when designing the network structure or similarity function. In Table 5, we analyze the deep learning methods for loop closure detection in this section. To compare the models and validation datasets utilized by different algorithms more intuitively, we summarize the models and validation datasets utilized in the integration of deep learning with VO and loop closure detection in Table 6.

It can be seen from Table 5 that most methods are supervised, and the research of loop closure detection has gradually moved towards outdoor scenes. Although various high-performance networks combined with other technologies can achieve better results, they also lead to slow inference speeds and long train times. Feature post-processing is a feasible solution, and designing a simplified similarity function may also be a good idea.

**Table 5.** A summary of the loop closure detection methods.

| Method | Supervision | Scenario | Contributions | Weakness |
|---|---|---|---|---|
| [75] | Unsupervised | Indoor | stacked autoencoder<br>combine denoising, sparsity, and continuity | network training is time-consuming |
| [30] | | Campus, Indoor, | stacked denoise autoencoder<br>discuss the effect of hyper-parameters | without any constraints |
| [76] | | Indoor, Outdoor | lightweight network<br>multi-view and illumination invariance<br>easy to deploy | unable to display keyframes that match the current frame |
| [78] | | Indoor | multi-scale feature fusion<br>weighted feature node | fewer comparison algorithms |
| [88] | Supervised | Urban, Campus | compress redundant feature information<br>temporal similarity constraint | it is not clear how to choose the compression ratio |
| [81] | | Outdoor | combines super dictionary, Bow dictionary and deep learning<br>ResNet50 FCN | only compare with traditional BoW |
| [83] | | Urban, Campus | two forward passes of a single network to extract global and local features<br>EfficientNetB0 | has not been tested in a complete system |
| [84] | | Urban | geometrical consistency check based on LDB descriptors<br>R50-DELG | poor performance with few landmarks |
| [85] | | Outdoor Urban | local motion and structure consensus (LMSC) | without a comparison of time consumption |

In Table 6, most methods are focused on monocular and RGB-D sensors; the KITTI and TUM RGB-D are the most commonly used datasets. The fewer the training datasets, the worse the generalization ability. This also explains the poor performance across datasets in some methods. The lightweight network architectures are more popular for ensuring real-time performance. Many methods prefer lightweight networks, such as MiniNet, DUNet and SegNet. In summary, various deep neural models have successfully employed different modules of a VSLAM system. Most existing methods are limited to specific datasets, and it is still necessary to add different types of datasets to improve the generalization ability of the models. Network parameters should be minimized as much as possible while maintaining precision to accelerate the inference speed.

**Table 6.** The models utilized in the combination of deep learning with VO and loop closure detection.

| Type | Method | Year | Sensor | Model | Dataset |
|---|---|---|---|---|---|
| feature extraction | [37] | 2018 | Monocular | MagicPoint + SuperPoint | HPatches |
| | [43] | | Monocular, RGB-D | dual-branch recurrent network | KITTI + ICL_NULM |
| | [40] | 2019 | RGB-D | geometric correspondence network (GCN) v2 | TUM RGB-D |
| | [45] | | | YOLOV1 + Mask-RCNN | |
| | [44] | | Monocular, RGB-D | Tfeat + hard negative mining strategy | TUM + HPatches |
| | [42] | 2021 | Monocular | learned invariant feature transform (LIFT) | KITTI + Euror |
| | [46] | | Monocular RGB-D | YOLOV3 | Stevens dataset |
| | [47] | | | Darknet19-YOLOv3 | |
| | [48] | 2022 | RGB-D | YOLOV4 | TUM RGB-D |
| | [49] | | | YOLOv5n | |
| | [50] | 2021 | Monocular, RGB-D | SalNavNet | saliency dataset + EuRoc |
| | [51] | 2017 | Monocular, RGB-D | dispnet | |
| | [53] | | Monocular, Binocular, Stereo | dispnet | KITTI |
| | [54] | 2018 | Monocular | DepthNet, PoseNet, ResFlowNet | |
| | [56] | | Monocular | GAN, convLSTM | |
| motion estimation | [57] | 2019 | | stacked GANs, LSTM | |
| | [58] | | Monocular, RGB-D | YOLOV2, MS-CNN | TUM + KITTI |
| | [55] | | Monocular | DepthNet, PoseNet | KITTI + EuRoc |
| | [61] | 2020 | RGB-D | DUNet | TUM RGB-D |
| | [60] | | | VIOLearner | KITTI |

**Table 6.** *Cont.*

| Type | Method | Year | Sensor | Model | Dataset |
|---|---|---|---|---|---|
| | [59] | 2021 | Monocular | Mask-RCNN | KITTI |
| | [63] | | Monocular, RGB-D, Stereo | Mask-RCNN | KITTI + TUM RGB-D |
| | [64] | 2022 | RGB-D | Mask-RCNN | TUM RGB-D |
| | [67] | 2016 | Monocular | vsLSTM, DPP | SumMe + TVSum |
| selection of keyframes | [66] | 2019 | Monocular | depth predictor, keyframe selector, camera motion estimator | KITTI |
| | [68] | 2019 | Monocular | MiniNet | KITTI |
| | [74] | 2021 | RGB-D | Mask-RCNN, SegNet | TUM RGB-D |
| | [75] | 2015 | RGB-D | stacked autoencoder | TUM RGB-D |
| | [30] | 2017 | Monocular, RGB-D | stacked denoise autoencoder | New College + City Center + TUM |
| loop closure detection | [76] | 2018 | | denoising autoencoder network | KITTI |
| | [78] | 2019 | | AlexNet | Matterport3D |
| | [88] | 2020 | Monocular | ResNet18 ConvNet | Gardens Point + Nordland + KITTI |
| | [81] | | | CNN, autoencoder | KITTI + City Center + Garden Point Walk |
| | [83] | 2021 | | EfficientNetB0 | KITTI |
| | [84] | 2022 | Monocular, RGB-D | ResNet50-FCN | KITTI + Malaga + Oxford |
| | [85] | | Monocular | R50-DELG | KITTI + New College |

## 5. Mapping with Deep Learning

Mapping is not a separate thread and requires VO, loop closure detection, and other methods to generate the environmental information needed for the map. Based on the acquired positional information, pixel depth, and semantic information, they are mapped to a 2D image or 3D point cloud to construct a map corresponding to the real environment. We divide the different map representations into geometric mapping, semantic mapping, and general mapping according to the different ways of scene expression.

### 5.1. Geometric Mapping

Geometric mapping mainly detects the shape and structure description of the scenes, and scene representation mainly includes the depth, voxel and mesh.

#### 5.1.1. Depth

Depth is the most common representation of understanding a scene's geometric and structural information. Accurate depth estimation contributes to the absolute scale recovery of VSLAM. The common practice is to formulate the depth estimate as a mapping function for the input image and train the deep neural network to predict the depth of each pixel in the source image through a large amount of data with depth labels.

However, this method requires a large amount of data and seriously reduces robustness and generalization in the absence of labeled data. In unsupervised research, research scholars have redefined depth prediction as a problem of view synthesis. Godard et al. [53] adopted photometric consistency as self-supervised signals and utilized disparity images to reconstruct the depth of each pixel. Zhou et al. [51] adopted temporal consistency as self-supervised signals and recovered the ego-motion while achieving depth estimation. Based on this framework, many efforts [54,55,66,89–94] have extended the network structure to various degrees and all obtained good results. With regard to loss functions, various constraints [90,92,93,95] are added to improve performance.

#### 5.1.2. Voxel

Voxel representation is the most common in 3D geometry and is similar to the pixel in the 2D image. SurfaceNet [96] is an end-to-end framework for multiview stereopsis, which can learn photo-consistency and geometric relations of the surface structure. To efficiently encode the camera parameters, a novel 3D voxel representation is proposed, namely the colored voxel cube (CVC). It takes two CVCs from different viewpoints as input and predicts the surface confidence of each voxel. The traditional depth fusion pipelines

may worsen the photo-consistency for a sparse environment with a large baseline angle. Ji et al. [97] proposed a fully convolutional network, SurfNet+, and introduced a novel occlusion-aware view selection approach and a multi-scale strategy for the incompleteness and inaccuracy problems introduced by sparse multi-view stereopsis. RayNet [98] combines CNN with Markov random fields (MRF) to reconstruct scenario geometry. CNN is used to extract view-invariant features and MRF can explicitly model perspective geometry, occlusion and other physical processes. Xie et al. [99] adopt ResNet18 and ResNet50 to design an encoder–decoder network framework, Pix2Vox++, which reconstructs 3D voxel grid representations from multiple RGB images. Additionally, a multi-scale context-aware fusion module is proposed to adaptively select high-quality reconstructions from rough 3D volumes in parallel.

The limitation of voxel representation is the huge computation which makes it hard to guarantee real-time performance in a high-resolution map. Tatarchenko et al. [100] proposed an efficient convolutional decoder architecture which can generate high resolution 3D outputs in an octree representation. Compared with standard decoders which perform on regular voxel grids, this method does not have cubic complexity and allows higher-resolution outputs with limited memory. Based on the encoder–decoder architecture, Henzler et al. [101] utilized CNN to predict volumetric representations in the voxel, learning 3D textures from 2D exemplars.

### 5.1.3. Mesh

The mesh representation encodes the 3D object geometry according to the underlying structure, including edges, vertices and faces. Pixel2Mesh [102] is an end-to-end deep learning framework that consists of an image feature network and a cascaded mesh deformation network. The cascaded mesh deformation network leverages the perceptual features extracted from input images to progressively deform an ellipsoid mesh into the 3D model. Scan2Mesh [103] takes scan point cloud data of 3D objects as input and combines the CNN and graph-based neural network to achieve one-to-one discrete mapping between the predicted and ground truth points and generate a compact mesh. However, both of these methods can only reconstruct single objects. Bloesch et al. [104] utilize 2.5D triangular meshes as the compact geometry representation of the scene geometry and predict the vertex coordinates of the planes directly from the source image, and the remaining vertex depth is optimized as a free variable.

The map constructed via geometric mapping is always sparse; geometric feature information, such as depth, edge and point, extracted by a deep neural network is easier to obtain. However, the single depth generally cannot be used for practical applications. Although the voxel incorporates many types of geometric information, it requires a large amount of computation and memory for a high-resolution map. Similarly, the mesh is limited to a few applications. Hence, it seems that the geometric map is not a good choice. Incorporating semantic information into maps, such as in semantic maps, may provide a more advantageous alternative for leveraging the capabilities of deep learning while enabling more user-friendly visualization.

### 5.2. Semantic Mapping

Semantic mapping not only contains scene geometric information but also incorporates higher-level semantic information. According to the different types of scene segmentation, the discussion is divided into three parts: semantic segmentation, instance segmentation and panoramic segmentation.

### 5.2.1. Semantic Segmentation

Semantic segmentation classifies image pixel points by associating a semantic label to each pixel point in the image. SemanticFusion [105] takes RGB-D images as input and obtains pixel-level semantic information from a deconvolutional semantic segmentation network; the dense 3D map is reconstructed by the ElasticFusion SLAM [106] system. Based

on SemanticFusion, Li et al. [107] proposed a 3D semantic map reconstruction method for road scenes. Inspired by the MobileNet [108], the pyramid scene parsing network is adopted to predict the pixel-level classification of keyframes and generate the optimal depth estimation for each keyframe. The depthwise separable convolution decreases the model size and computation, and the pyramid pooling [109] combines multi-scale feature maps as global contextual priors. The semantic information of keyframes can be fused into a globally consistent 3D semantic map which is reconstructed based on the correspondence between labeled pixels and voxels in the 3D point cloud. L. Ma et al. [110] proposed a self-supervised deep neural network to predict multi-view consistent semantics from RGB-D images and generate the consistent semantic labels of a map by adding different constraints. DA-RNN [111] proposed a novel recurrent neural network architecture for semantic labeling in RGB-D videos in which a novel data association recurrent unit is introduced to capture dependencies between frames, and DA-RNN is combined with KinectFusion [112] to accomplish 3D semantic scene reconstruction.

For stereo images, STDyn-SLAM [113] utilizes a stereo camera to capture stereo image pairs as well as depth images and employs a SegNet [114] network architecture similar to that of VGG16 to perform the semantic segmentation of dynamic objects on the left image. The local point cloud which is constructed via the combination of the depth map, the semantic result of the left image and VO information is updated into the global point cloud to construct an octomap eventually. This work can run in the indoor or outdoor environment in real-time at a fast speed, but it is still challenging for large-scale scenarios.

### 5.2.2. Instance Segmentation

While semantic segmentation can assign a category to each pixel, the objects with the same label cannot distinguish different individuals. When we need information about individuals, then instance segmentation is required, which can separate out specific individuals in a category. Compared to semantic segmentation, instance segmentation does not need to label each pixel, but only needs to find the edges or contours of the object.

Fusion++ [115] adopts Mask-RCNN to provide instance masks that are fused into the TSDF reconstruction. It generates a long-term map that concentrates on the most significant objects with variable, object size-dependent resolution, remaining good performance even in realistic cluttered scenes. MaskFusion [116] also adopts Mask-RCNN, but it is able to achieve non-rigid and dynamic scene reconstructions which are not considered by Fusion++. Sünderhauf et al. [117] maintain individual objects as the key entities in the map and combine bounding box-based object detection with unsupervised 3D segmentation to generate an object-level semantic map. Grinvald et al. [118] proposed a novel method that can incrementally generate object-level volumetric maps and discover novel objects that have not been seen in the scene. The 3D shape, location and semantic information are incrementally fused into the global map using a map integration strategy.

### 5.2.3. Panoptic Segmentation

Unlike instance segmentation which is only concerned with objects, panorama segmentation needs to detect all the objects in the scene, including the background. On the level of stuff and things, PanopticFusion [119] proposed a volumetric semantic mapping system that combines pixel-wise panoptic label prediction with volumetric integration and is able to achieve dense semantic labeling, large-scale 3D reconstruction and labeled mesh extraction while distinguishing individual objects. AVP-SLAM [120] leverages powerful semantic features to construct 3D maps and locate vehicles in parking lots. Four surround view cameras are employed to increase the perception range, and semantic features are extracted using a modified Unet, including guide signs, stop lines, speed bumps, etc. These features are projected as global coordinates to construct a global semantic map that can be used to achieve localization at the centimeter level with good robustness and precision even in dark underground parking lots. Panoptic MOPE [121] introduced a fast semantic segmentation and registration weight prediction convolutional neural network (Fast-RGBD-

SSWP) to achieve panoptic segmentation, combining geometric, appearance, and semantic information in the registration cost function with adaptive weights. This system can not only generate high-quality panoptic reconstructions but also output the 6D pose of objects in the scene.

The advantages of semantic map are that they can better utilize deep learning technology and provide more user-friendly visualization. They can include more semantic information and achieve more accurate object detection and recognition. In addition, semantic maps can use less memory and computing resources, making them more efficient in practical applications. One of the disadvantages is that their creation requires high-quality semantic annotations to ensure map accuracy and usability. Additionally, they may overly focus on semantic information of objects or scenes, while neglecting their geometric features, which may lead to suboptimal results for some applications. There are still some challenges to be solved, such as how to integrate multi-source and multi-modal data and improve the map updating speed and real-time performance.

### 5.3. General Mapping

Geometric and semantic mapping can be considered forms of explicit map representation, and encoding the whole scene as an implicit representation using deep learning models is called general representation, which cannot be intuitively understood by humans but contains important scene information required for the task. A map can be represented as an implicit representation through deep autoencoders, neural rendering models, neural radiance fields, etc.

#### 5.3.1. Deep Autoencoder

The deep autoencoders compress the map and encode the high-dimensional data as a high-level compact representation. CodeSLAM [122] generates a compact and optimizable representation of dense scene geometry by training a variational autoencoder network on the intensity map. Because of its limited size, both camera poses and depth maps can be jointly estimated for overlapping keyframes to obtain global consistency. Based on CodeSLAM, CodeMapping [123] extends the variational autoencoder by introducing factor-graph-based optimization of DeepFactors [124] to further improve global consistency and accuracy. This variational autoencoder is easy to be integrated into a keyframe-based SLAM system without delaying the main system thread. Park et al. [125] introduced a continuous signed distance function (SDF) representation, DeepSDF. It utilized a continuous volumetric field to represent continuous implicit surfaces of shape and the 3D shapes are learned by probabilistic autoencoders.

#### 5.3.2. Neural Rendering Model

Neural rendering can improve rendering quality and efficiency by learning 3D scene information from training data. Specifically, neural rendering uses neural network models to convert 3D scenes into 2D pixel values to achieve more realistic visual effects. Neural rendering has a wide range of applications in virtual reality, game development, and film production. GQN [126] encodes scene images acquired from different viewpoints into a scene representation, and utilizes this representation to predict the appearance of the current scene from a new viewpoint. The introduction of a geometric-aware attention mechanism extends the system to accomplish more complex environment modeling. Sitzmann et al. [127] proposed a continuous, 3D structure-aware scene representation, the scene representation network (SRN), which implicitly represents the scene as a continuous differentiable function that maps global coordinates to a feature representation of local scene attributes. It has greater potential for joint shape and appearance interpolation, view synthesis, etc. Lombardi et al. [128] learned the 3D volume latent representation of dynamic scenes using the coding–decoding network architecture and incorporated the implicit surface-based representation into its framework.

### 5.3.3. Neural Radiance Field

Neural radiance fields emerged in recent years, and is the first continuous neural scene representation. It utilizes deep neural networks to model the volumetric density and color of a 3D scene. NeRF [129] utilizes the multi-layer perceptron (MLP) to represent a continuous scene as a 5D vector-valued function that adopts a 3D location and 2D viewing direction as input and outputs volume density and directional emitted radiance at any point in space. To further improve performance and speed, positional encoding and the hierarchical sampling procedure are leveraged to map the input location to a higher dimensional space and to sample the high frequency information more efficiently during training process, respectively, which helps MLP learn high-frequency information better and obtain more detailed scene representations. Although this continuous compact representation shows great advantages over discrete representations such as grids and voxels, it requires more memory space as the scene complexity increases and takes longer to optimize for each new scene.

Based on NeRF, generative radiance fields (GRAF) [130] introduce generative adversarial networks to generate a variant scene representation with 3D consistency. However, this work is limited to simple single-object scenarios. For it to be applied in more complex scenarios with multiple objects, GIRAFFE [131] improves GRAF. Unlike NeRF and GRAF, it outputs the 3D color of points in space; what GIRAFFE outputs are abstract features, which is why its scene representation is called generative neural feature fields. An affine transformation is introduced for each object and background in the scene to control the pose, shape and appearance of individual objects. The existing problems are disentanglement failure and dataset bias. Many subsequent efforts [132–138] have improved the NeRF, for example, Pi-GAN [132] proposed a variant NeRF with the SIREN periodic activation function. ShapeGAN [133] not only considers different views, but also takes into account different illumination conditions.

One of the limitations of autoencoder-generated compact representations is their lack of visibility. In contrast, neural rendering can generate high-quality images with high levels of photorealism and accurately simulate complex lighting and material properties. However, it does not always generalize well to new scenarios, and the quality of output images is directly dependent on the quality and quantity of training data available. NeRF is a recent development in the field of neural rendering; it similarly requires substantial computational resources and careful training to produce accurate results. A classification of existing approaches on deep learning-based mapping can be found in Table 7.

**Table 7.** A classification of mapping methods.

| Mapping Type | Map Representation | Method |
|---|---|---|
| Geometric Mapping | Depth | [51,53–55,66,89–95] |
| | Voxel | [96–101] |
| | Mesh | [102–104] |
| Semantic Mapping | Semantic Segmentation | [105,107,111,113] |
| | Instance Segmentation | [115–118] |
| | Panoptic Segmentation | [119–121] |
| General Mapping | Deep Autoencoder | [122,123,125] |
| | Neural Rendering Model | [126–128] |
| | Neural Radiance Field | [129–138] |

## 6. Datasets and Evaluation Metric

### 6.1. Datasets

In the early stages, many SLAM datasets are generated by the LiDAR. However, the image information is more significant to VSLAM, and the different types of images are processed in different ways because of their properties, including monocular, binocular, stereo.

The KITTI was created jointly by the Karlsruhe Institute of Technology (KIT) in Germany and Toyota Technological University Chicago (TTIC), mainly contains downtown,

highway, countryside and other scenes. It not only consists of monocular, binocular, and stereo 1240 × 370 image sequences, but also an inertial measurement unit (IMU) and GPS data. It is very popular in visual odometry, 3D object detection and 3D tracking.

EuRoC is collected by a micro air vehicle (MAV) and published at the European Robotics Challenge. It contains mainly binocular data and IMU data from two scenarios: a large industrial environment and a room in ETH Zurich.

The TUM RGB-D dataset was released by the Technical University of Munich and captured data through RGB-D sensors. It consists of two different indoor scenes with a total of 39 sequences, providing 6D pose ground truth labels. At the same time, it proposes a kind of evaluation criteria and indicators.

Oxford RobotCar contains over 20 million images taken by six vehicle-mounted cameras, as well as information collected by laser ranging, GPS, and inertial navigation; it is commonly used in the positioning and mapping of autonomous vehicles in dynamic outdoor environments. It almost covers all periods, weather and scene changes, including various scenes (such as pedestrians, vehicles, and road construction), different weather conditions (such as heavy rain, light rain, and snow) and different periods (such as direct sunlight, night and dusk). Table 8 summarizes some popular datasets for VSLAM. We also arrange the link of each dataset for readers' convenience.

**Table 8.** Overview of VSLAM datasets.

| Dataset | Year | Type | Link |
|---------|------|------|------|
| KITTI [139] | 2012 | Monocular, Binocular, Stereo | http://www.cvlibs.net/datasets/kitti/index.php accessed on on 23 May 2023 |
| Oxford RobotCar [140] | 2016 | Monocular, Stereo | https://robotcar-dataset.robots.ox.ac.uk/ accessed on on 23 May 2023 |
| EuRoC [141] | 2016 | Monocular, Stereo | https://projects.asl.ethz.ch/datasets/doku.php?id=kmavvisualinertialdatasets accessed on on 23 May 2023 |
| TartanAir [142] | 2020 | Monocular, Stereo | http://theairlab.org/tartanair-dataset/ accessed on on 23 May 2023 |
| MVSEC [143] | 2018 | Binocular, Stereo | https://daniilidis-group.github.io/mvsec/ accessed on on 23 May 2023 |
| Complex Urban [144] | 2019 | Binocular, Stereo | https://www.complexurban.com/ accessed on on 23 May 2023 |
| Málaga Urban [145] | 2014 | Stereo | https://www.mrpt.org/MalagaUrbanDataset accessed on on 23 May 2023 |
| Cityscapes [146] | 2016 | Stereo | https://www.cityscapes-dataset.com/ accessed on on 23 May 2023 |
| Apollo | 2018 | Stereo | https://apollo.auto/synthetic.html accessed on on 23 May 2023 |
| Rosario [147] | 2019 | Stereo | https://www.cifasis-conicet.gov.ar/robot/doku.php accessed on on 23 May 2023 |
| FinnForest [148] | 2020 | Stereo | https://etsin.fairdata.fi/dataset/06926f4b-b36a-4d6e-873c-aa3e7d84ab49 accessed on on 23 May 2023 |
| DSEC [149] | 2021 | Stereo | https://dsec.ifi.uzh.ch/ accessed on on 23 May 2023 |
| InteriorNet [150] | 2018 | Stereo, RGB-D | https://interiornet.org/ accessed on on 23 May 2023 |
| RGB-D Object [151] | 2011 | RGB-D | http://rgbd-dataset.cs.washington.edu/ accessed on on 23 May 2023 |
| TUM RGB-D [152] | 2012 | RGB-D | https://vision.in.tum.de/data/datasets/rgbd-dataset/download accessed on on 23 May 2023 |
| NYUV2 [153] | 2012 | RGB-D | https://cs.nyu.edu/~silberman/datasets/nyu_depth_v2.html accessed on on 23 May 2023 |

**Table 8.** *Cont.*

| Dataset | Year | Type | Link |
|---|---|---|---|
| ScanNet [154] | 2017 | RGB-D | http://www.scan-net.org/ accessed on on 23 May 2023 |
| ETH3D-SLAM [155] | 2019 | RGB-D | https://www.eth3d.net/slam_datasets accessed on on 23 May 2023 |
| Newer College [156] | 2020 | Binocular, RGB-D | https://ori-drs.github.io/newer-college-dataset/ accessed on on 23 May 2023 |
| OpenLoris-Scene [157] | 2020 | Binocular, RGB-D | https://lifelong-robotic-vision.github.io/dataset/scene accessed on on 23 May 2023 |

*6.2. Evaluation Metric*

We can approach the evaluation of VSLAM system performances from different perspectives, such as time consumption, complexity, precision, robustness and so on. The popular evaluation metrics used to evaluate trajectory precision are absolute trajectory error (ATE) and relative pose error (RPE). Additionally, a lightweight evaluation tool, EVO, can help researchers conveniently evaluate system performance. Additionally, precision and recall are usually used to evaluate loop closure detection. The relationship between true positive (TP), false negative (FN), false positive (FP) and true negative (TN) is shown in Table 9's confusion matrix. The details of evaluation metrics can be found in Table 10.

**Table 9.** Confusion matrix.

| Truth \\ Prediction | Loop Detected | Not Detected |
|---|---|---|
| Loop detected | TP | FN |
| not detected | FP | TN |

**Table 10.** Evaluation metric.

| Metric | Alias | Definition | Description |
|---|---|---|---|
| Trajectory Precision | ATE | $\sqrt{\frac{1}{N}\sum_{i=1}^{N}\left\|\log_e\left(T_{g,i}^{-1} * T_{esti,i}\right)^V\right\|_2^2}$ | $N$ represents the number of points taken; $T_{g,i}$ represents the Euclidean transformation of the i-th point of the truth trajectory $T_{est,i}$ represents the Euclidean transformation of the i-th point of the estimation trajectory |
| | ATE (trans) | $\sqrt{\frac{1}{N}\sum_{i=1}^{N}\left\|trans\left(T_{g,i}^{-1} * T_{esti,i}\right)\right\|_2^2}$ | ATE consists of a translation part and rotation part trans means the translation part of the error |
| | RPE | $\sqrt{\frac{1}{N-\Delta}\sum_{i=1}^{N-\Delta}\left\|\log_e\left(T_{g,i}^{-1} * T_{g,i+\Delta}\right)^{-1}\left(T_{esti,i}^{-1} * T_{esti,i+\Delta}\right)^V\right\|_2^2}$ | $\Delta$ indicates the time period between the points taken RPE consists of a translation part and rotation part |
| Loop Closure Metric | P | $\frac{TP}{TP+FP}$ | Precision and recall are paradoxical. In the VSLAM system, we put more emphasis on precision and compromise the recall a little. |
| | R | $\frac{TP}{TP+FN}$ | |

Alias: ATE represents absolute trajectory error; RPE represents relative pose error; P represents precision; R represents recall.

## 7. Open Problems and Future Directions

The deep learning-based VSLAM methods have undergone great progress in recent decades, but there are still many unresolved issues, and researchers still face various challenges.

*7.1. Open Problems*

1.  Data Association

RGB-D cameras provide depth information and IMU provides motion measurement information. The integration of these data provided by different sensors can facilitate the

performance of VSLAM systems. Almost all existing VSLAM systems are jointly optimized using multiple sensors, which can better handle complex scenes and environments. However, the perceptual data of different sensors, with different data types, value ranges, coordinate systems and time stamps, need to be processed uniformly before using these data. Otherwise, it is easy to increase the amount of system operations and affect the efficiency and real-time performance. The more sensors, the more data to be processed. How to process different sensor data is also a problem worth discussing.

2.  Uncertainty

Differently from human-labeled datasets, there are various unexpected situations and uncertain factors in the real world. For instance, these include serious motion blur in camera images caused by the robot shaking violently because of the uneven road; the alternating light and dark environment affects the camera exposure during imaging. These uncertainties may affect the robustness of the system and make the whole VLSAM system fail.

3.  Application Scenarios

The evaluation of the existing systems is limited to some specific scenarios, such as underground parking, city roads, and so on. For instance, the neural rendering models and NeRF heavily rely on the quality of the dataset. This also means that a whole new network needs to be trained for a new scenario that has been never met before, which is very resource-intensive in practical applications. How to improve the generalization of the network models to perform VSLAM tasks in different scenarios remains an open problem.

4.  Interpretability

Despite the fact that the deep learning boom has promoted the development of different fields, deep learning has been long-criticized as a "black box", and the greatest limitation is its poor interpretability, which makes some scholars reluctant to use deep learning methods and prefer traditional methods with rigorous mathematical formula reasoning and clear correspondence. Some research on interpretability has emerged in recent years. Zhang et al. [158] focus on visual interpretability, such as feature visualization, and heat mapping. Adadi et al. [159] explain the interpretability of whole AI methods, including neural networks and other AI models [160]. Additionally, Fan et al. [161] cite more than 200 articles to make a comprehensive and detailed summary. Interpretability is crucial to the development of deep learning. Once a breakthrough is made, it will inevitably have a huge impact on the field of AI.

5.  Evaluation system

In Section 6.2 of this paper, it is revealed that ATE and RPE are widely used to evaluate the trajectory precision of a VSLAM system. To a certain extent, this is also a reflection of pose accuracy. Precision and recall are universally accepted to judge the quality of loop closure detection. However, there are no any reliable evaluation metrics for mapping, keyframes and features. For an extended period of time, subjective assessment has been the primary method for evaluating map quality. This approach is not sustainable in the long term, and more robust and objective evaluation methodologies are necessary to ensure accuracy and reliability. Therefore, the assessment of the mapping effect and the accuracy of selected keyframes and extracted features remains an unresolved challenge.

*7.2. Future Directions*

Although the VSLAM technologies still confront diverse issues, they pose both challenges and opportunities. Inspired by the other reviews of VSLAM, we discuss the following prospects and hope that our review will inspire new research efforts in these important fields.

1.  New Sensors

The most common sensors include IMU, cameras, GPS and LiDAR. New sensors, such as event cameras [162], millimeter-wave radar [163], thermo cameras [164] and magnetic sensors [159], with more accurate and robust perceptual data, can increase precision and

finish tasks better. The combination of multiple sensors allows complementary advantages. However, the unusual sensors are still under-researched. Multi-sensor fusion is inevitable, and it is believed that more new sensors will also emerge.

2. Map Representation

We have summarized many map representations in this paper, such as the geometric map, semantic map and general map. Most of the existing map representations are designed in advance, and a unified map representation for different tasks and environments may cause a waste of resources and an increase in computation. To construct a VSLAM system, one can autonomously choose the most appropriate map type and construct a map that can meet task requirements is a research hotspot in the future.

3. Lifelong Learning

Most deep learning works obtain the pre-train models based on a closed-form dataset, and the pre-train models are retrained or fine-tuned on the actual dataset. In the actual open world, the changing external environment and various unpredictable factors need to be confronted, which also requires deep learning models to continuously learn to adapt to the complicated and changeable world.

4. Multi-robot Cooperation

With the research of VSLAM gradually turning to the outdoor large-scale environment, multi-robot cooperation can act and perceive independently in a large-scale environment at the same time, which can effectively improve the efficiency and stability of the VSLAM system and increase the speed of tasks, such as search, rescue, planetary exploration and military operations. However, the key problem that arises is how to combine the perceptual data of each robot into a unified framework; for example, how to piece together the local maps of each robot into a global map and accurately locate them. Although there have been numerous research attempts with multi-robot cooperation [165–168], many problems are still unresolved.

5. Semantic VSLAM

Semantic VSLAM combines the VSLAM system with semantic information through deep learning to improve the scene understanding of robots and the intelligence of human–robot interaction. In Section 5.2 of this paper, the integration of semantic information and mapping is the most prominent one. Compared with the point cloud map in which the points cannot be understood by humans, the semantic map can show various objects on the map more intuitively and achieve a higher level of scene understanding, which is of great significance for the intelligence of VSLAM. The combination of semantic and VSLAM can achieve complementary advantages and make good progress in object recognition, target detection, semantic segmentation, and semantic mapping.

6. Lightweight and Miniaturization

Limited by the hardware conditions of some platforms and the requirement of actual tasks, the architecture of various deep learning neural networks increasingly demands the property of being lightweight, and the equipment applications are becoming more and more miniaturized. This paper has shown that many efforts focus on lightweight network [44,47] and hardware deployment optimization [39]. In some practical application scenarios, miniature search and rescue robots in natural disasters through the narrow and hazardous region, and micro-UAVs reconnoiter the theater of operations in military operations and other applications of VSLAM on a mobile phone and embedded system. These small devices generally can neither carry huge sensors and computers with powerful computing ability, nor can they afford too much computation, requiring the VSLAM system to be lightweight and miniaturized.

In summary, the application of deep learning technology to VSLAM has yielded promising results, and the incorporation of semantic information achieves a more advanced understanding of a scene. However, the interpretability of deep learning and the limitation

of datasets can be a significant obstacle to the development of VSLAM. To ensure real-time performance and practicality, lightweight networks and miniature mobile devices are required. On the other hand, the research of VSLAM develops towards large-scale environments. Although multi-robot cooperation seems to be a feasible solution, the processing of data from multiple robots, multi-target localization and global map construction have become new challenges. Undoubtedly, multiple sensors' fusion and multi-robot cooperation will become research hotspots in future development.

## 8. Conclusions

Although the research of deep learning-based VSLAM has started late, its amazing development rate shows its great potential. In this paper, we emphasize the combination of deep learning and VSLAM from three aspects: VO, loop closure detection, and mapping. One or more modules of VSLAM are replaced with deep learning-based methods, such as feature extraction, depth estimation, and 3D reconstruction, which greatly improve precision and efficiency. The introduction of semantic information not only allows the system to obtain richer feature information, but also makes it easier to achieve high-level human–robot interaction and scene understanding. The integration of VSLAM and deep learning has effectively addressed certain limitations of traditional VSLAM approaches. However, new issues have arisen, such as the explainability of deep learning, the huge network architecture and the reliability of datasets. To help the researchers in this filed work efficiently, we provide the links for each widely used dataset. The widely used evaluation metrics are also summarized.

As VSLAM research transitions from the restricted indoors to the expansive outdoors, relying on a single sensor to accomplish SLAM becomes increasingly challenging. The integration of multiple sensors is essential to achieve reliable and precise SLAM performance. Researchers gradually tend to explore solutions to multi-sensor fusion and large-scale environments. Deep learning-based VSLAM not only provides a data-driven alternative, but also expands our thinking of next-generation AI spatial perception. We believe that the interpretability of deep learning will be realized in the near future, and the whole VSLAM system will be achieved in deep learning methods. We hope that this paper can contribute to the development of VSLAM technology.

**Author Contributions:** Conceptualization, Y.Z. and Y.W.; methodology, Y.Z. and Y.W.; software, H.C.; validation, K.T. and Y.Y.; formal analysis, Y.Z. and Y.Y.; investigation, Y.Z.; resources, H.C.; data curation, Y.Z., H.C. and K.T.; writing—original draft preparation, Y.Z.; writing—review and editing, Y.Z., Y.W. and H.C.; visualization, K.T.; supervision, Y.W.; project administration, Y.Z., Y.W. and Y.Y.; funding acquisition, Y.W. All authors have read and agreed to the published version of the manuscript.

**Funding:** This work was funded by the National Natural Science Foundation of China (grant number 61573183) and the Open Project Program of the National Laboratory of Pattern Recognition (NLPR) (grant number 201900029). These funds come from Y.W.

**Data Availability Statement:** Not applicable.

**Acknowledgments:** Research in this article was supported by the National Natural Science Foundation of China (grant number 61573183) and the Open Project Program of the National Laboratory of Pattern Recognition (NLPR) (grant number 201900029) are deeply appreciated. The authors would like to express our sincere gratitude to the reviewers and editors for their careful evaluation and valuable suggestions on our manuscript. Their insightful comments and guidance have greatly contributed to the improvement of our research. We greatly appreciate their recognition and support for our work.

**Conflicts of Interest:** The authors declare no conflict of interest.

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
