# Peer review of "Review of Visual Simultaneous Localization and Mapping Based on Deep Learning"

_remotesensing, doi:10.3390/rs15112740_

Round 1

Reviewer 1 Report

This paper systematically reviews VSLAM methods based on deep learning. It then briefly reviews the development history of VSLAM and introduces its basic principles and framework. We then focus on the combination of deep learning and VSLAM in three areas: visual odometer (VO), closed-loop detection, and mapping.But, obviously, this is more of a working paper than a review. Does not contain enough summary and framework. This article still needs considerable revision.

 Extensive editing of English language required.

Reviewer 2 Report

The paper is well written, it summarizes a good number of techniques and the authors did use a good amount of updated and relevant references.

Introduction does provide context and highlights the paper contribution.

 However, the paper needs a more detailed discussion in section 7.

- I suggest to add a general discussion that gives an overview of the content/findings of the paper;

- When discussing "open problems and future directions", please link the idea/discussion with the previous sections of the paper. It is a long paper and it is not easy for the reader to understand why exactly the authors concluded something.   

In conclusion, I suggest to link the conclusion with the paper objective and its contribution.

Minor Suggestion

- To change to bullet lists instead of numbered list along the text, or maybe using a level 3 sections (as in section 5).

Reviewer 3 Report

remotesensing-2371895

In this paper, the authors review the VSLAM methods based on deep learning. Further, briefly review the history of VSLAM development and introduce its fundamental principles and framework. Then, focus on the combination of deep learning and VSLAM from three inspects: visual odometry (VO), loop closure detection, and mapping. The author’s work is timely under consideration but to properly reach the publication, I have some suggestions that must be considered in revision.

1.     Figure 1 images are very blur. Please add clear and more visible images.

2.     Please add a full stop at the end of every caption.

3.     There are numerous grammatical mistakes and typos that must be corrected with a detailed proofreading of the overall manuscript.

4.     There are several monotonous words, please avoid it.

5.     Moreover, the authors have cited every figure and table at the end of the paragraph in a separate sentence which seems very strange and unpleasant. Such as “Table 1 shows the comparison between different map types, and Figure 1 shows the visualization of different maps.”, “Table 2 summarizes and analyses above deep learning methods on feature extraction.” It is suggested to cite inside the paragraph discussion.

6.     To know more about SLAM techniques, the authors may refer to “, “Simultaneous Localization and Mapping based on Kalman Filter and Extended Kalman Filter,” doi.org/10.1155/2020/2138643,”.

7.     A detailed summary table of all the existing approaches dealing with Visual Simultaneous Localization and Mapping should be added which clearly summarize the existing work and the gap still exist.

8.     As this is a review paper, therefore, the authors should add more challenges and their possible solutions.

9.     Some recent references should be added, especially from the last 2 years. 

 There are numerous grammatical mistakes and typos that must be corrected with a detailed proofreading of the overall manuscript.

Round 2

Reviewer 1 Report

Thanks to the author's efforts, my problem has been further explained.

However, there are still some problems which I think need to be considered.

First, some summaries are added to the article. But I think this kind of summary is relatively simple, or a pile of results. I think it would be more clear if in each section, for the different methods, the physical characteristics of the methods and the suitability of the objectives were given.

It has to be said that this paper has done a lot of preliminary work, and the language expression has made some progress, but it still lacks an effective explanation of the principle of the target and remains on the surface of the problem. I suggest that the table can be improved to combine the same elements and make a comprehensive analysis and explanation behind the table.

Moderate editing of English language.

Reviewer 3 Report

No more comments from side. Authors have already addressed all my comments. 

NA

Author Response

We would like to express our sincere gratitude to the reviewers for their careful evaluation and valuable suggestions on our manuscript. Their insightful comments and guidance have greatly contributed to the improvement of our research. We greatly appreciate their recognition and support for our work.
